# STRUCTURE LEARNING FOR UNFAITHFUL DISTRIBUTIONS: THE MINIMAL DEPENDENCE FAITHFULNESS

## ABSTRACT

Causality detection is to identify the "true" directed acyclic graph (DAG) of a causal model from the joint probability distribution of the observed variables. Algorithms such as PC and its modified versions perform this task under the restrictive faithfulness assumption, that is the DAG encodes all conditional independencies imposed by the distribution. However, all existing algorithms fail to detect the simple structure where a variable is the XOR of several Bernoulli variables, violating faithfulness. We generalize this type of unfaithfulness that appears in other, non-XOR, examples and define the *minimal dependence* of a given variable $X$ as the set of variables, such that $X$ is independent of each variable in the set but depends on at least one of them, the *dependent member* if conditioned on the remainder of the set. Minimal dependencies of size at least two violate faithfulness. Consequently, we relax faithfulness to *minimal dependence faithfulness*, restricting the neighbors of a node to its dependent members, and impose *minimal orientation faithfulness* that generalizes the orientation rules under faithfulness. We then determine the structure of the dependent members of a node $X$ in the true DAG and show that they are connected to $X$ either directly or indirectly by a collider. Finally, we provide a sound and complete modification of the PC algorithm to detect this kind of unfaithfulness and output all possible candidates for the true DAG.

## 1 INTRODUCTION

Structural causal models (SCMs) are used to capture the causal relationships between a set of random variables. They assign to each variable a deterministic function of some of the other variables, known as its *parents* or *direct causes*, and possibly a noise variable (Peters et al., 2017). Consequently, each SCM induces a joint probability distribution over the random variables and a directed acyclic graph (DAG) where the direct causes of each variable are linked to that variable. The DAG is also known as the "true DAG" or *causal network* of the variables. To learn the structure of the causal network – a process referred to as *structure learning* – intervention is a reliable method to find causal relations (Ke et al., 2020). However, because of practical constraints, the use of only observational data has attracted attention in recent decades (Ng et al., 2021). For this purpose, sufficiency, causal Markov condition (CMC), and faithfulness are usually considered in the literature to make learning possible (Zhang & Spirtes, 2016). Causal sufficiency refers to the absence of hidden (or latent) variables. CMC implies that all conditional independencies implied by the true DAG hold in the joint probability distribution and faithfulness implies the inverse.

Under these assumptions, one main approach to find the true DAG is known as *constraint-based* and is based on testing conditional independencies (constraints) between the variables (Kitson et al., 2021). Examples of constraint-based algorithms are PC (Spirtes et al., 2000), SGS (Spirtes et al., 2000) FCI (Guo et al., 2020) which are based on (in)dependence detection. The result of this approach is a class of independence-equivalence (I-equivalence) graphs that are presented as a partially DAG (PDAG). Under the causal Markov and faithfulness assumptions, constraint-based approaches have been shown to correctly find the true PDAG given the joint probability distribution, and the same holds asymptotically with respect to the number of data instances of a given dataset (Ng et al., 2021).

CMC and faithfulness are used to conclude the existence and absence of links in the true DAG based on conditional independence test results, respectively. However, the faithfulness assumption is violated in some practical situations (Andersen, 2013), making it a restrictive assumption. Some

of the work in the literature has accordingly relaxed this assumption (Lemeire et al., 2012) and developed *unfaithfulness detection* approaches, that is, to detect from observational distribution $P$ the existence of some conditional dependencies that cannot be captured by the true DAG (Ramsey et al., 2006; Zhang & Spirtes, 2008; Spirtes & Zhang, 2014).

Adjacency-faithfulness and orientation-faithfulness assumptions that are weaker than faithfulness were defined in (Ramsey et al., 2006). PC algorithm was accordingly adapted, resulting in the *conservative PC (CPC) algorithm* that puts forward a number of candidate PDAGs that include the true DAG. As another relaxation of faithfulness, triangle-faithfulness was introduced and investigated in (Zhang & Spirtes, 2008; Spirtes & Zhang, 2014), resulting in the *very conservative PC (VCPC)* and *VCSGS* algorithms. However, all of these and other existing algorithms, such as MGM-FCI-MAX (Horii, 2021), fail to detect the true graph in examples such as the generalized XOR where several Bernoulli random variables cause another variable.

The first step in detecting this type of unfaithfulness was taken in (Marx et al., 2021) for the case with three variables, i.e., two causing the third, resulting in the conventional XOR example. The authors relaxed the faithfulness to the so-called 2-adjacency faithfulness and 2-orientation faithfulness, provided a modified grow-shrink (GS) algorithm (Margaritis & Thrun, 1999) to detect triple of variables violating faithfulness, and proved the soundness.

We take the next step in this regard and define the *minimal dependence* of a given variable $X$ as the set of variables $Y$, such that $X$ is independent of each $Y$ but becomes dependent if conditioned on the remainder of the set. Minimal dependencies of size at least two violate faithfulness. Consequently, we relax the assumption to *minimal dependence faithfulness*, limiting the existence of links to a variable and its minimal dependence. We then determine the structure of minimal dependencies in the true DAG and provide a modification of the PC algorithm, the minimal dependence PC (MD-PC) algorithm to detect this kind of unfaithfulness and output all possible candidates for the true DAG.

## 2 PROBLEM FORMULATION

Consider a *structural causal model (SCM)* (Peters et al., 2017) defined as the tuple $\langle \mathcal{X}, \mathcal{N}, P_\mathcal{U}, \mathcal{F} \rangle$ where $\mathcal{X} = \{X_1, \ldots, X_m\}$ are the random variables, $\mathcal{N} = \{N_X \mid X \in \mathcal{X}\}$ are the disjoint noise random variables whose joint distribution $P_\mathcal{U}$ satisfies $P_\mathcal{U}(\mathcal{N}) = \prod_X P_\mathcal{U}(N_X)$. The value assigned to each variable $X \in \mathcal{X}$ is a deterministic function $f_X$ of a subset of the variables $\mathrm{Pa}_X \subset \mathcal{X} \setminus \{X\}$, known as its *parents*, and a parent noise variable $N_X \in \mathcal{N}$, i.e., $X := f_X(\mathrm{Pa}_X, N_X)$. The set of functions $f_X$ for all variables $X$ defines $\mathcal{F}$. The parents of each variable $X$ are known as its *direct causes*. The function $f_X$ is *minimal* in the parents of $X$; that is, there does not exist another function $g_X$, such that $f_X(\mathrm{Pa}_X, N_X) = g_X(\mathcal{S}, N_X)$ for some subset $\mathcal{S} \subset \mathrm{Pa}_X$. The SCM induces *(i)* a joint (observational) probability distribution $P$ over the variables $\mathcal{X}$ and *(ii)* a DAG $\mathcal{G}$ whose nodes represent variables $\mathcal{X}$, and there is a link to every variable $X$ from each of its parents $\mathrm{Pa}_X$.

The goal is to obtain DAG $\mathcal{G}$, referred to as the "true DAG," from observational probability distribution $P$, that is to estimate the causal relationships from the observational distribution.

**Problem 1 (Causal discovery)** *Consider an SCM over the variables $\mathcal{X}$, inducing DAG $\mathcal{G}$ and joint observational probability distribution $P$. Given distribution $P$, find DAG $\mathcal{G}$.*

This is, however, an impossible task in general. The simplest example is the case with two dependent variables $X$ and $Y$, where both SCMs $X := 2Y$ and $Y := 0.5X$ impose the same observational distribution $P$ but two different DAGs $Y \to X$ and $X \to Y$.

Some commonly-imposed assumptions facilitate this task. One is the *(causal) faithfulness* assumption explained as follows. True DAG $\mathcal{G}$ implies some independencies between the variables. For example, if there is no trail (path) between two variables in $\mathcal{G}$, their deterministic functions are not related, rendering the two variables independent. Faithfulness states that such independencies implied by $\mathcal{G}$ capture all independencies satisfied by the distribution $P$. The attribution of (conditional) independencies to a DAG is formalized by the notion of *d-separation* (see Appendix). Let $\mathcal{I}(\mathcal{G})$ denote the set of all d-separations in a DAG $\mathcal{G}$ and $\mathcal{I}(P)$ denote the set of all conditional independencies implied by distribution $P$, i.e., $\mathcal{I}(P) = \{(\mathcal{Y}_1 \perp \mathcal{Y}_2 \mid \mathcal{Y}_3) : \mathcal{Y}_1, \mathcal{Y}_2, \mathcal{Y}_3 \subseteq \mathcal{X}\}$.

**Assumption 1 (Faithfulness)** $\mathcal{I}(P) \subseteq \mathcal{I}(\mathcal{G})$.

Distribution $P$ is said to be *(causally) faithful* to the DAG $\mathcal{G}$ if it satisfies the above assumption. Problem 1 can then be stated as follows.

**Problem 2 (Causal discovery under faithfulness)** *Consider an SCM over variables $\mathcal{X}$, inducing DAG $\mathcal{G}$ and joint observational probability distribution $P$. Given distribution $P$ and under Assumption 1, find DAG $\mathcal{G}$.*

Problem 2 is often introduced from a different perspective, without the causality ingredient: there is some true DAG $\mathcal{G}$ over random variables $\mathcal{X}$ with joint distribution $P$, and the goal is to again obtain $\mathcal{G}$ from $P$. Clearly, then the connection between $\mathcal{G}$ and $P$ is lost and any DAG is a candidate for $\mathcal{G}$. Faithfulness helps to bridge the gap yet is insufficient as, for example, an empty graph satisfies the faithfulness assumption for every distribution $P$, and hence, is always a valid candidate. A second commonly-imposed assumption is the *(Causal) Markov Condition (CMC)*.

**Assumption 2 (Markovness)** $\mathcal{I}(\mathcal{G}) \subseteq \mathcal{I}(P)$.

If Markovness holds, $\mathcal{G}$ is called an *I-map* for $P$. If additionally, $\mathcal{G}$ is no longer an I-map for $P$ upon the elimination of an edge, then it is a *minimal I-map* for $P$. Similar to faithfulness, Markovness is insufficient to learn causal DAGs from observational data, because, for example, a fully connected DAG has $\mathcal{I}(\mathcal{G}) = \emptyset$ and satisfies CMC for any observational distribution. However, together these assumptions enable us to present Problem 2 without the causality ingredient.

**Problem 3 (Structure learning)** *Consider random variables $\mathcal{X}$ with joint probability distribution $P$. Assume there exists a DAG $\mathcal{G}$ satisfying Assumptions 1 and 2. Given distribution $P$, find DAG $\mathcal{G}$.*

We state the results in the conventional setup of Problem 3, but they can also be stated using the setup of Problem 1. Note that even with both Markovness and faithfulness in force, the two-node DAGs example stated at the beginning of this section may not be distinguished using the observational distribution. This is because the two have identical distributions $P$ and in turn $\mathcal{I}(P)$. Consequently, the problem of finding the true DAG is reduced to finding the so-called *I-equivalent* class of DAGs, which share the same $\mathcal{I}(G)$ that is equal to $\mathcal{I}(P)$. I-equivalent DAGs are known to have the same skeleton and same directions on some of the edges (Koller & Friedman, 2009), and hence, are represented by a PDAG.

PC algorithm solves Problem 3 (and 2) by putting forward an I-equivalence class that includes the true DAG $\mathcal{G}$. The algorithm, however, fails if the faithfulness assumption is violated. The idea with PC is that if two variables are independent conditioned on any subset of the other variables, then the two are not adjacent in the true DAG–a condition satisfied by faithfulness. If distribution $P$ is unfaithful, PC may incorrectly identify the absence of a link once an independence is detected in $P$ as illustrated in the following example.

**Example 1 ($n^{\text{th}}$-order XOR)** *Consider the SCM with variables $Y_1, \ldots, Y_n$, and $X$ defined by*

$$Y_i := \text{Bernoulli}(1/2), \quad i = 1, \ldots, n,$$
$$X := Y_1 \oplus \ldots \oplus Y_n,$$

*where $\oplus$ is the XOR function. The induced causal DAG is shown in Figure 1(a). By probabilistic arithmetic, $P(X \mid Y_i) = P(X)$ for all $i$. Therefore, $X$ is independent from every $Y_i$, but $X$ is not independent of any $Y_i$ conditioned on the remainder of $Y_i$'s, i.e., $P(X \mid Y_1, \ldots, Y_n) \neq P(X \mid Y_i)$ for all $i$, implying the existence of some path between $X$ and $Y_1, \ldots, Y_n$. Therefore, the independencies $X \perp Y_i$ for all $i$ are satisfied by the distribution $P$ but not by the causal DAG $\mathcal{G}$. As a result, faithfulness is violated, i.e., $\mathcal{I}(P) \nsubseteq \mathcal{I}(\mathcal{G})$.*

*Should existing constraint-based algorithms, i.e., SGS and PC or their modified versions such as CSGS, CPC, VCSGS, or VSPC, be applied to the above observational distribution, the result would be the empty DAG $\mathcal{G}'$ in Figure 1(b), which is different from $\mathcal{G}$ and violates the causal Markov condition, i.e., $(X \perp Y_1 \mid Y_2, \ldots, Y_n) \in \mathcal{I}(\mathcal{G}')$ whereas $(X \perp Y_1 \mid Y_2, \ldots, Y_n) \notin \mathcal{I}(P)$.*

To the best of our knowledge, except for the case of $n = 2$ (Marx et al., 2021), this type of unfaithfulness has not been investigated in detail and no constraint-based algorithm is able to detect it. The unfaithfulness is also not limited to Bernoulli distributions, binary variables, or deterministic relationships; see Examples 4 and 5 in the Appendix. It is also not limited to extended v-structures as

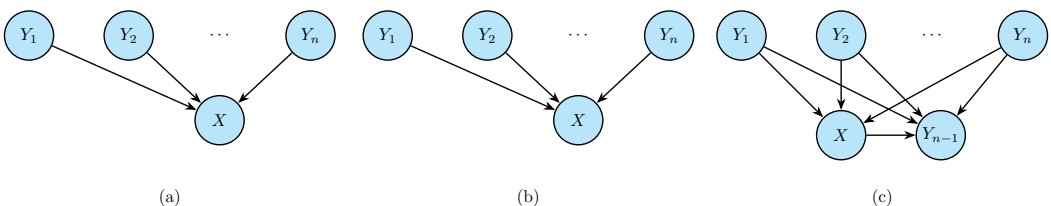

Figure 1: (a) The true DAG in Example 1, (b) the DAG found by PC and its modified versions, (c) the true DAG in Example 2

in Figure 1(a) as illustrated in the following example. Given set of variables $\mathcal{Y} = \{Y_1, \ldots, Y_n\}, n \geq 1$, define $\mathcal{Y}_{-i} = \mathcal{Y} \setminus \{Y_i\}$ for $i = 1, \ldots, n$.

**Example 2** *Consider the SCM with variables $Y_1, \ldots, Y_n$, and $X$ defined by*

$$Y_i := \text{Bernoulli}(1/2), \quad i = 1, \ldots, n-1,$$
$$X := Y_1 \oplus \ldots \oplus Y_{n-1} \oplus U,$$
$$Y_n := Y_1 \oplus \ldots \oplus Y_{n-1} \oplus X,$$

*where $U := \text{Bernoulli}(1/2)$ is the noise variable. The induced causal DAG $\mathcal{G}$ is shown in Figure 1(c). For every index $i$, it can be verified that $X \perp Y_i$, but $X \not\perp Y_i \mid \mathcal{Y}_{-i}$. Thus, similar to Example 1, the induced joint probability distribution $P$ is unfaithful.*

In what follows, we first characterize the type of unfaithfulness appearing in these examples, then determine the structure of the underlying true DAG, and finally propose an algorithm that identifies a class of DAGs that includes the true DAG.

## 3 MINIMAL-DEPENDENCE FAITHFULNESS

The type of unfaithfulness explained in Examples 1 and 2 happens when a variable $X$ is independent of each individual member of a set $\mathcal{Y} = \{Y_1, \ldots, Y_n\}$ but not when conditioned on the other members. Namely, $X$ indeed depends on all of $Y_1, \ldots, Y_n$, yet each single $Y_i$ does not individually inform the distribution of $X$. The following definition captures this idea.

**Definition 1 (Weak minimal dependence)** *Given a variable $X \in \mathcal{X}$, a minimal dependence for $X$ is a set $\mathcal{Y} \subseteq \mathcal{X} \setminus \{X\}$ that satisfies the following conditions:*

 *1. there exists at least one member $Y \in \mathcal{Y}$ that satisfies*

$$X \not\perp Y \mid \mathcal{Y} \setminus \{Y\}, \tag{1}$$

 *2. if $|\mathcal{Y}| \geq 2$, then*

$$\forall Y \in \mathcal{Y} \quad X \perp Y, \tag{2}$$

 *3. no proper subset of $\mathcal{Y}$ satisfies (1) and (2).*

If a minimal dependence of a variable $X$ consists of a single variable $Y$, then Condition (2) does not apply. Condition (1), on the other hand, implies that $X$ and $Y$ are dependent. Should faithfulness be in force, we would conclude that $X$ and $Y$ are connected by some active trail.

However, if a minimal dependence entails at least two variables, it violates the faithfulness assumption. Condition (1) implies that $X$ depends on some variable $Y$ in its minimal dependence set given the remainder of the set. However, Condition (2) implies that $X$ is independent of every member of the minimal independence. This is counter intuitive as it implies that the only way the dependence between $X$ and $Y$ can be explained is by the other members of the minimal dependence; nevertheless, $X$ is independent of each of these members, resulting in an unfaithful distribution.

**Proposition 1** *Consider random variables $\mathcal{X}$ with joint probability distribution $P$. If for any of the variables $X \in \mathcal{X}$, there exists a minimal dependence of $X$ with cardinality of at least two, then $P$ is not faithful to any DAG $\mathcal{G}$ over $\mathcal{X}$ that is an I-map for $P$.*

The set $\mathcal{Y} = \{Y_1, \ldots, Y_n\}$ in Examples 1 and 2 is a dependence set for $X$. This notion of unfaithfulness may as well apply to some real-world scenarios, where the presence of a single "effect" does not make a particular "cause" more likely, yet the combination of several such effects may do so.

**Example 3 (influenza or dengue fever)** *Let the binary variable $X$ indicate the presence of either influenza ($X = 0$) or the dengue fever disease (Halstead, 2007; Eccles, 2005) ($X = 1$) in a patient. There are three symptoms shared between these two diseases:* (i) *fever (captured by $Y_1 = 1$ when present otherwise $Y_1 = 0$);* (ii) *headache (captured by $Y_2 = 1$ when present otherwise $Y_2 = 0$);* (iii) *joint and muscle pain (captured by $Y_3 = 1$ when present otherwise $Y_3 = 0$). Knowing any of these symptoms does not make any of the two diseases more likely, i.e., $P(X \mid Y_i) = P(X)$ for all $i$. However, while all three symptoms can be simultaneously present in the case of influenza, that is not the case with dengue fever. In dengue fever, fever and headache appear in the early stages, and the joint and muscle pain is observed often a few days after. Therefore, if the patient is known to have fever and headache, the additional information of whether they also have joint and muscle pain may determine the disease type, i.e., $P(X \mid Y_1, Y_2, Y_3) \neq P(X \mid Y_1, Y_2)$, implying $X \not\perp Y_3 \mid Y_1, Y_2$. It follows that $\mathcal{Y} = \{Y_1, Y_2, Y_3\}$ is a minimal dependence for $X$.*

The third condition in Definition 1 is to ensure that the members of $\mathcal{Y}$ are all required for the unfaithfulness; otherwise, the dependence set $\mathcal{Y}$ may become redundantly large by including variables that are independent of every other variable, e.g., isolated nodes. Nevertheless, the consequence of minimality is that all members of the minimal dependence set satisfy Condition 1.

**Proposition 2** *Let $\mathcal{Y} \subseteq \mathcal{X}$ be a weak minimal dependence of a variable $X \in \mathcal{X}$. Then*

$$\forall Y \in \mathcal{Y} \quad X \not\perp Y \mid \mathcal{Y} \setminus \{Y\} \tag{3}$$
$$X \not\perp \mathcal{Y} \tag{4}$$

*and for every $\mathcal{Y}' \subset \mathcal{Y}$,*

$$X \perp \mathcal{Y}', \tag{5}$$
$$\forall Y \in \mathcal{Y}' \quad X \perp Y \mid \mathcal{Y}' \setminus \{Y\}. \tag{6}$$

Although we defined minimal dependence in Definition 1 in its simplest form by requiring the dependence in Condition (1) to hold for only one member, it appears that the condition does hold for all members of the minimal dependence. This means that in practice, if for a set of variables $\mathcal{Y} = \{Y_1, \ldots, Y_n\}$ that are each independent of $X$, we find that just one of them, say $Y_1$, depends on $X$ conditioned on the rest of the variables, then so do all of the remainder, i.e., $Y_2, \ldots, Y_n$, provided that $\mathcal{Y}$ is minimal. If $\mathcal{Y}$ is not minimal, then we can find the minimal set by finding the smallest subset of $\mathcal{Y}$ that satisfies Condition (1). This explains why a uniform quantifier is not used in Condition (1).

### 3.1 THE STRUCTURE OF THE MINIMAL DEPENDENCE

What can be concluded about the true DAG $\mathcal{G}$ of a distribution $P$ that admits a minimal dependence of size at least two? An immediate result of Proposition 1 is that once a minimal dependence of size at least two is detected for a variable $X$ in the distribution $P$, the absence of the links between $X$ and its minimal dependence variables may no longer be concluded, even though $X$ is independent of them. This, however, does not mean that $X$ and the minimal dependence can form any DAG. The following result enforces the existence of certain combinations of links.

**Theorem 1** *Consider random variables $\mathcal{X}$ with joint probability distribution $P$. Consider a node $Z \in \mathcal{X}$ and its weak minimal dependence $\mathcal{Y}$. If DAG $\mathcal{G}$ is an I-map for $P$, then every member $Y \in \mathcal{Y}$ is connected to $Z$ either by a collider-free trail or by a trail, every collider of which has a node in $\mathcal{Y}$ as its descendant.*

To further conclude about the structure of the true DAG, we define the following stronger version of Definition 1. This definition is motivated by normal faithfulness, where two nodes $X$ and $Y$ are adjacent if and only if their dependence does not break upon the observation of any other set $\mathcal{U}$, i.e., $X \not\perp Y \mid \mathcal{U}$ for all $\mathcal{U} \not\supseteq \{X, Y\}$ (Koller & Friedman, 2009). Similarly, here, we extend Condition (1) in Definition 1 to when any additional set $\mathcal{U}$ is observed.

**Definition 2 ((Strong) minimal dependence)** *Given a variable $X \in \mathcal{X}$, a* (strong) minimal depen-
dence *for $X$ is a set $\mathcal{Y} \subseteq \mathcal{X} \setminus \{X\}$ that satisfies the following conditions:*

    *1. there exists at least one member $Y \in \mathcal{Y}$ that satisfies*

$$\forall \mathcal{U} \in \mathcal{X} \setminus (\{X\} \cup \mathcal{Y}) \qquad X \not\perp Y \mid (\mathcal{Y} \setminus \{Y\}) \cup \mathcal{U} \tag{7}$$

    *2. if $|\mathcal{Y}| \geq 2$, then*

$$\forall Y \in \mathcal{Y} \quad X \perp Y, \tag{8}$$

    *3. no proper subset of $\mathcal{Y}$ satisfies* (7) *and* (8).

*Every member $Y \in \mathcal{Y}$ that satisfies* (7) *is called a* dependent *member of $\mathcal{Y}$ and the set of all dependent
members of $\mathcal{Y}$ is denoted by $\mathcal{Y}^{\mathrm{o}}$. The set of all strong minimal dependencies of $X$ is denoted by*
$\mathbf{Dep}(X)$.

In the rest of the paper, by "minimal dependence" we mean strong minimal dependence. According
to Definition 2, the dependence between $X$ and its minimal dependence variables does not break
upon the observation of non-minimal-dependence variables. Clearly, a strong minimal dependence of
a variable $X$ is also its weak minimal dependence but not the other way around. Note that (7) implies
the presence of an edge between $X$ and a node in $\mathcal{Y}$.

Next, to deduce the absence of an edge in the true DAG, we make the following assumption. The
violation of faithfulness no longer allows edge elimination in $\mathcal{G}$. Nevertheless, faithfulness is only
violated "within" the minimal independence sets. The outer ones may be removed.

**Assumption 3 (Minimal-dependence faithfulness)** *Consider DAG $\mathcal{G}$ over variables $\mathcal{X}$ with joint
probability distribution $P$. If $Y, Z \in \mathcal{X}$ are adjacent in $\mathcal{G}$, then each is a dependent member of a
minimal dependence of the other, i.e., there exists a minimal dependence $\mathcal{Y} \in \mathbf{Dep}(Z)$ such that
$Y \in \mathcal{Y}^{\mathrm{o}}$ and a minimal dependence $\mathcal{Z} \in \mathbf{Dep}(Y)$ such that $Z \in \mathcal{Z}^{\mathrm{o}}$.*

This assumption is motivated by a result in faithfulness (as well as adjacency faithfulness (Ramsey
et al., 2006); see Section 3.4). Under a faithful distribution, two nodes $Y$ and $Z$ are adjacent only if
$Y \not\perp Z \mid \mathcal{U}$ for all $\mathcal{U} \subseteq \mathcal{X} \setminus \{Y, Z\}$. Namely, adjacency is allowed only between "tightly" dependent
variables. Similarly, Assumption 3 enforces the same, with the difference that here the notion of
tight dependence extends to that between a variable $Z$ and a **set** of other variables $\mathcal{Y}$. Namely, in
view of Proposition 2, each minimal dependence $\mathcal{Y} \in \mathbf{Dep}(Z)$ can be seen as a "super node":
Although independent of individual nodes in $\mathcal{Y}$, variable $Z$ does depend on the whole $\mathcal{Y}$ according
to Equation (7). Hence, Assumption 3 allows adjacency between nodes $Z$ and $Y$ only if $Z$ and $\mathcal{Y}$
are tightly dependent where $Y \in \mathcal{Y}^{\mathrm{o}}$ and $\mathcal{Y} \in \mathbf{Dep}(Z)$ which in turn implies that that $Z$ and $Y$ are
tightly dependent.

Assumption 3 and Definition 2 sharpen Theorem 1 to what we define as a v-star. Denote the union
of the minimal dependencies of $Z$ by $\overline{\mathbf{Dep}}(Z) = \cup_{\mathcal{Y} \in \mathbf{Dep}(Z)} \mathcal{Y}$ and the union of the sets of the
dependent members of the minimal dependencies by $\mathbf{Dep}^{\mathrm{o}}(Z) = \cup_{\mathcal{Y} \in \mathbf{Dep}(Z)} \mathcal{Y}^{\mathrm{o}}$.

**Definition 3 (v-star)** *A* v-star *over the node set $\mathcal{X}$ is the DAG where a single node $X \in \mathcal{X}$ is
connected to every other node in $\mathcal{X}$ either directly (i.e., the two are adjacent) or indirectly by a
collider whose effect node is also a node in $\mathcal{X}$. The DAG is also referred to as a* v-star centered at $X$.

**Theorem 2** *Consider random variables $\mathcal{X}$ with joint probability distribution $P$, and let $Z \in \mathcal{X}$. If
DAG $\mathcal{G}$ is an I-map for $P$ and satisfies Assumption 3, then* (i) *$Z$ and $\mathbf{Dep}^{\mathrm{o}}(Z)$ form a v-star centered
at $Z$; and* (ii) *for every $Y \in \mathbf{Dep}^{\mathrm{o}}(Z)$ that is not adjacent to $Z$, a node in $\mathcal{Y} \setminus \{Y\}$ is a descendant
of (or equals) the effect node of the collider connecting $Y$ to $Z$, where $\mathcal{Y}$ is the minimal dependence
of $Z$ that includes $Y$.*

The structures in Examples 1 and 2 both appear to be v-stars. Theorem 2 does not reveal the structure
of each individual minimal dependence but all of them as a whole: Dependent member $Y$ of a
minimal dependence $\mathcal{Y} \in \mathbf{Dep}(Z)$ may be connected to $Z$ directly, indirectly by another node in $\mathcal{Y}$,
or indirectly by another node in a different minimal dependence of $Z$, say $\mathcal{Y}'$. However, this requires
that other node to belong to the minimal dependence of $Y$ as otherwise, there is no link between

them according to Assumption 3. So if none of the members of the minimal dependencies belong to any of the other minimal dependencies of $Z$, then Theorem 2 is further sharpened to each minimal dependence of $Z$ forming a v-star. Another special case is for singleton minimal dependencies as illustrated by the following corollary.

**Corollary 1** *Consider random variables $\mathcal{X}$ with joint probability distribution $P$, and let $Z \in \mathcal{X}$. If DAG $\mathcal{G}$ is an I-map for $P$ and satisfies Assumption 3, then for every singleton strong minimal dependence $\{Y\} \in \mathbf{Dep}(Z)$ (that is, for every $Y$ and $Z$ where $Y \not\perp Z \mid \mathcal{U}$ for all $U \subseteq \mathcal{X} \setminus \{X, Y\}$), it holds that $Z$ is adjacent to $Y$.*

### 3.2 THE SYMMETRIC CASE

So far, Theorems 1 and 2 indicate the possibility of intertwined connections of the minimal dependencies in the true DAG. The connections can be partly separated upon the introduction of some symmetry to the minimal dependence.

**Definition 4 (Symmetric minimal dependence)** *Given variables $\mathcal{X}$ with joint probability distribution $P$, a* symmetric minimal dependence *is a set of random variables $\mathcal{Z} = \{Z_1, \ldots, Z_n\} \subseteq \mathcal{X}$, such that for every $i \in \{1, \ldots, n\}$, $\mathcal{Z}_{-i} \in \mathbf{Dep}(Z_i)$ and $\mathcal{Z}_{-i}^{\mathrm{o}} = \mathcal{Z}_{-i}$.*

Namely, for each variable of a symmetric minimal dependence, the remainder of the set is its strong minimal dependence and all members of which are dependent members. The set of variables in both Examples 1 and 2 are symmetric minimal dependencies, but not those in Example 4. What structure does the true DAG of a symmetric minimal dependence take? It appears that in both examples, there is a node where all others are directing to. We show that this is generally true.

**Definition 5 (Directed star)** *A* directed star *over nodes $\mathcal{Z}$ consists of a single* center *node $Z \in \mathcal{X}$ that has an incoming edge from every other node $\mathcal{Z} \setminus \{Z\}$. The directed star is also referred to as a star directed at $Z$.*

**Corollary 2** *Consider random variables $\mathcal{X}$ with joint probability distribution $P$. If DAG $\mathcal{G}$ is an I-map for $P$ and satisfies Assumption 3, then the nodes of a symmetric minimal dependence admit a directed star in $\mathcal{G}$.*

Unlike the general results in Theorems 1 and 2 on the structure of the union of the minimal dependencies, here, the structure of a single minimal dependence is revealed. Namely, if a node $X$ and its minimal dependence form a symmetric minimal dependence, their graphical representation includes a directed star, regardless of the rest of the nodes in $\overline{\mathbf{Dep}}(X)$. The center node, however, is undetermined. For example, for a symmetric minimal dependence of size three, the possible DAGs include all cases in Figure 2-a–c) as well as Figure 2-d–f) where an additional link is included. Thus, there can be several candidate DAGs for the true DAG, which are not distinguishable based on the distribution $P$. This motivates defining a class for such candidate DAGs as in the following section.

**Remark 1** *A special case of a symmetric minimal dependence is a set of two variables $X$ and $Y$, where $X \not\perp Y \mid \mathcal{U}$ for all $\mathcal{U} \in \mathcal{X} \setminus \{X, Y\}$. Then Corollary 2 implies that there is a link between $X$ and $Y$ in the true DAG. Therefore, as highlighted in Corollary 1, every node that forms a singleton minimal dependence of $X$ is adjacent to $X$.*

### 3.3 MINIMAL DEPENDENCE EQUIVALENCE CLASS

The idea with PDAG class P-maps is to capture all DAGs that are an I-map for the distribution $P$, satisfy the faithfulness condition, and are indistinguishable with respect to the distribution $P$. Similarly, here, we can define a class of DAGs that all are an I-map for the distribution $P$ and satisfy minimal dependence faithfulness, and that potentially any of them can be the true DAG.

So far, neither the definition of a minimal dependence nor the minimal dependence faithfulness assumption helps to determine the direction of the edges of the true DAG. This is because faithfulness does not hold and hence an independence between a triple of variables does not necessarily imply an immorality between them in the true DAG. More specifically, we build upon the following result for faithful distributions that helps in finding the orientation of undirected edges found in constraint-based

algorithms (Koller & Friedman, 2009). Suppose that in a P-map, nodes $Y_1$ and $Y_2$ are not adjacent but are both adjacent with $X$. Then

1. if $Y_1, Y_2, X$ form the immorality $Y_1 \to X \leftarrow Y_2$ (that is $Y_1$ and $Y_2$ are not adjacent and both are linked to $X$), then $Y_1 \not\perp Y_2 \mid X, \mathcal{U}$ for all $\mathcal{U} \subseteq \mathcal{X} \setminus \{X, Y_1, Y_2\}$;

2. if $Y_1, Y_2, X$ do not form an immorality, then $Y_1 \not\perp Y_2 \mid \mathcal{U}$ for all $\mathcal{U} \subseteq \mathcal{X} \setminus \{X, Y_1, Y_2\}$.

Now, in our case, instead of the single nodes $Y_1$ and $Y_2$, we have the minimal dependencies $\mathcal{Y}_1$ and $\mathcal{Y}_2$ that can be viewed as "super nodes" and that form directed stars centered at $X$. Accordingly, the above conditions are extended by replacing $Y_i$ with $\mathcal{Y}_i$ and making necessary adjustments. First, we extend the definition of immorality so that it also applies to node sets.

**Definition 6** *Node $X$ and node sets $\mathcal{Y}_1, \mathcal{Y}_2$ form an* immorality *if* (i) *there is no node in $\mathcal{Y}_1$ that is adjacent to a node in $\mathcal{Y}_2$ and* (ii) *$\mathcal{Y}_1 \cup \mathcal{Y}_2 \cup \{X\}$ forms a directed star centered at $X$.*

**Assumption 4 (Minimal orientation)** *Consider an SCM over variables $\mathcal{X}$, inducing DAG $\mathcal{G}$ and joint probability distribution $P$. For $X \in \mathcal{X}$, if $\mathcal{Y}_1 \cup \{X\}$ and $\mathcal{Y}_2 \cup \{X\}$ are symmetric minimal dependencies, where no node in $\mathcal{Y}_1$ is adjacent with a node in $\mathcal{Y}_2$, then*

1. *if $\mathcal{Y}_1 \cup \mathcal{Y}_2 \cup \{X\}$ form an immorality in $\mathcal{G}$, then $\mathcal{Y}_1 \not\perp \mathcal{Y}_2 \mid X, \mathcal{U}$ for all $\mathcal{U} \subseteq \mathcal{X} \setminus (\{X\} \cup \mathcal{Y}_1 \cup \mathcal{Y}_2)$;*

2. *if $\mathcal{Y}_1 \cup \mathcal{Y}_2 \cup \{X\}$ do not form an immorality in $\mathcal{G}$, then $\mathcal{Y}_1 \not\perp \mathcal{Y}_2 \mid \mathcal{U}$ for all $\mathcal{U} \subseteq \mathcal{X} \setminus (\{X\} \cup \mathcal{Y}_1 \cup \mathcal{Y}_2)$.*

The main use of the above assumption is the contra-position of the second case. For symmetric minimal dependencies $\mathcal{Y}_1 \cup \{X\}$ and $\mathcal{Y}_2 \cup \{X\}$ where $\mathcal{Y}_1$ and $\mathcal{Y}_2$ are not adjacent, if $\mathcal{Y}_1 \perp \mathcal{Y}_2 \mid \mathcal{U}$ for some $\mathcal{U} \subseteq \mathcal{X} \setminus (\{X\} \cup \mathcal{Y}_1 \cup \mathcal{Y}_2)$, then $\mathcal{Y}_1 \cup \mathcal{Y}_2 \cup \{X\}$ form a directed start centered at $X$.

**Definition 7 (MD-equivalence)** *Given variables $\mathcal{X}$ with joint probability distribution $P$, the* MD-equivalence class *of $P$ is the set of every DAG that* (i) *satisfies Assumptions 3 and 4, and* (ii) *for all $Z \in \mathcal{X}$, $\mathrm{Dep}^\mathrm{o}(Z)$ admits a v-star centered at $Z$. Any two DAGs in this class are said to be* MD-equivalent *(with respect to $P$).*

The six DAGs in Figure 2-a–h) are MD-equivalent. One could additionally require every symmetric minimal dependence to admit a directed star in $\mathcal{G}$ in order for it to belong to the MD-equivalence class; however, that would be an (indirect) implication of Condition ii) in Definition 7 as Corollary 2 is a result of Theorem 2.

To simplify the representation of the MD-equivalence class, we define the following extended type of PDAGs. A *dashed PDAG* over vertices $\mathcal{X}$ is a tuple $(\mathcal{X}, \mathcal{E}_1, \mathcal{E}_2, \mathcal{E}_3)$, where $\mathcal{E}_1 \in \mathcal{X} \times \mathcal{X}$ is the set of directed edges, and $\mathcal{E}_2, \mathcal{E}_3 \subseteq \{\{X_1, X_2\} \mid X_1, X_2 \in \mathcal{X}, X_1 \neq X_2\}$ are the set of undirected and dashed edges, respectively, and that no pair of nodes appears in more than one edge set (i.e., the directions in $\mathcal{E}_1$ ignored, the edge sets are mutually exclusive).

**Definition 8 (Class dashed PDAG)** *The* (MD-equivalent) class dashed PDAG *of distribution $P$ over variables $\mathcal{X}$ is a dashed PDAG $\mathcal{G}$ over $\mathcal{X}$ where* (i) *there is no edge (of any form) between $X$ and $Y$ in $\mathcal{G}$ iff that is the case with all DAGs in the MD-equivalence class of $P$;* (ii) *there is a directed edge $X \to Y$ in $\mathcal{G}$ iff $X \to Y$ exists in all DAGs in the MD-equivalence class of $P$;* (iii) *there is an undirected edge $X - Y$ in $\mathcal{G}$ iff among the DAGs in the MD-equivalence class of $P$, some (at least one) include the link $X \to Y$ and the remainder (at least one) include the link $Y \to X$;* iv) *there is a dashed edge $X - -Y$ in $\mathcal{G}$ iff none of the previous cases hold.*

Back to Figure 2, the MD-equivalent class dashed PDAG of the class is represented by Figure 2-h). As another example, given the true DAG in Figure 3-a) in Example 7, the class dashed PDAG representation is shown in Figure 3-c). Examples of the MD-equivalent class in this case include Figure 4-a–d). How to obtain the possible MD-equivalent DAGs from the class dashed PDAG? Similar to the P-map class PDAG, one can turn undirected (un-dashed and dashed) edges into directed links as long as directed cycles and new immoralities do not appear. For example, in Example 7, a DAG with the immorality $Y_5 \to X \leftarrow Y_6$ would not belong to the class dashed PDAG as this would

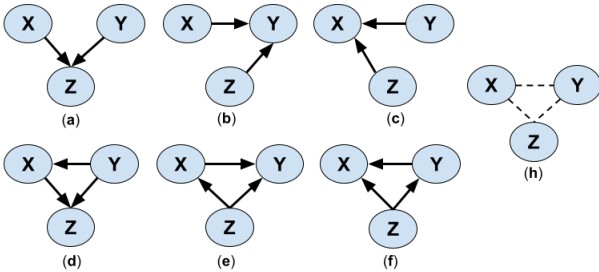

Figure 2: **a, b, c)** Three possible structures of an immorality over nodes $X, Y, Z$. **d, e, f)** Three possible structures of a triangle over nodes $X, Y, Z$. **h)** The graph representation for the existence of the immorality between nodes $X, Y, Z$.

cause the new immorality $Y_5 \rightarrow X \leftarrow Y_1$. The main difference to P-map class PDAG is that here the dashed edges can be removed as long as Condition *(ii)* in definition 8 is met. Again in Example 7, the case where all three dashed edges among $X, Y_1, Y_5$ are removed does not belong to the MD equivalent class.

### 3.4 RELATION TO OTHER TYPES OF FAITHFULNESS

The authors of (Ramsey et al., 2006) relaxed the faithfulness assumption to its implication in terms of the existence of a link. More specifically, they assume *adjacency faithfulness*, imposing that if nodes $X$ and $Y$ are adjacent in the true DAG $\mathcal{G}$, then they are dependent conditional on any subset of $\mathcal{X} \setminus \{X, Y\}$. Similarly, DAG $\mathcal{G}$ is said to be *minimally Markovian* if it is an I-map for $P$ and two nodes are adjacent in $\mathcal{G}$ if and only if they are dependent conditioned on every subset of other variables (Sadeghi & Soo, 2022). Both are the special case of Assumption 3 when all minimal dependencies are of size one. Moreover, the case where a symmetric minimal dependence is of size two matches the definition of tripe unfaithfulness in (Marx et al., 2021) and hence is considered as a special case. In this regard, Assumptions 3 and 4 and Definition 1 and 2 can be considered as extensions of (2-)adjacency and (2-)orientation faithfulness and the notion of $k$-*association* in (Marx et al., 2021), respectively. Given all this, if a DAG is faithful, it is adjacency faithful. If a DAG is adjacency faithful, it is 2-adjacency faithful, and if it is 2-adjacency faithful it is minimal-dependence faithful. So minimal-dependence faithfulness can be considered as a generalization of the adjacency and 2-adjacency faithfulness, and in turn, faithfulness. This, however, comes with the inevitable cost that the set of final candidates for the true DAG increases.

We highlight that the minimal dependence assumption imposed in this paper does not impose minimality on the number of edges and hence is different from existing notions such as *causal minimality* also known as *SGS minimality* stating that no proper sub-DAG of the true DAG $\mathcal{G}$ is an I-map for $P$ (Spirtes et al., 2000; Zhang & Spirtes, 2008; Neapolitan, 2004). It is also different from *P-minimality* (Zhang, 2013), stating that no DAG $\mathcal{G}'$ satisfying $\mathcal{I}(\mathcal{G}) \subset \mathcal{I}(\mathcal{G}')$ is an I-map for $P$. Another different notion is the *sparsest Markov representation (SMR)* which is an I-map $\mathcal{G}$, such that every other I-map that is not I-equivalent to $\mathcal{G}$ has more edges than $\mathcal{G}$ (Raskutti & Uhler, 2018). Similarly, an I-map $\mathcal{G}$ satisfying the *frugality* assumption, then it is in the set of SMRs (Forster et al., 2018). Overall, Assumption 3 does not imply any of the aforementioned minimality assumptions: the DAG in Figure 4-a) satisfies Assumption 3 but the sub-DAG in Figure 3-a) is the true DAG. Whether the aforementioned minimality assumptions imply Assumption 3 remains an open problem.

## 4 MD-PC ALGORITHM

How to perform causal discovery in the face of the unfaithfulness caused by a minimal dependence of size greater than one? A simple way is to iteratively find the **Dep** sets of all of the variables. However, the PC algorithm can be exploited to reduce the computations. To this end, we develop Algorithm 1, the *minimal dependence PC (MD-PC) algorithm*. Roughly speaking, the idea is to consider each minimal dependence as a super node and perform the PC algorithm as before. This determines the inter-structure of the minimal dependencies (super nodes). The intra-structure of the

minimal dependencies are then partially determined by Theorem 2 and Corollary 2. This applies only to minimal dependencies of size at least two, resulting in an unfaithfulness, detected according to Proposition 1. The result will be a set of candidate dashed PDAGs, one of which is guaranteed to be the true DAG under minimal-dependence faithfulness, i.e., Assumption 1.

Similar to PC, the algorithm goes through every pair of nodes $X$ and $Y$. If the two are dependent, then $Y$ does not belong to a minimal dependence of $X$ of size greater than one, and hence, normal PC is followed. Namely, if later, the two become conditionally independent, the connecting link is deleted; otherwise, $Y$ is adjacent with $X$ (equivalently, $\{Y\} \in \mathbf{Dep}(X)$). However, if $X$ and $Y$ are independent, we investigate the possibility of $Y$ forming a strong minimal dependence of size at least two with some other nodes. If yes, the minimal dependence is stored and all links between $X$ and the minimal dependence are removed. Otherwise, the link between $X$ and $Y$ is removed. Next, we proceed to mark the minimal dependencies using the minimal orientation rule.

**Theorem 3** *Consider an SCM over variables $\mathcal{X}$, inducing DAG $\mathcal{G}$ and joint probability distribution $P$. If $\mathcal{G}$ satisfies Assumptions 3 and 4, then Algorithm 1 outputs the MD-equivalent class dashed PDAG of $P$.*

**Remark 1.** Compared to PC, the MD-PC algorithm additionally computes $\mathbf{Dep}(X)$ for all $X$. Although this is not done in a separate process and uses the results of PC, it can result in additional computational burden. However, the computational complexity of the proposed MD-PC and the PC algorithm both equal $\mathcal{O}(2^N)$, where $N$ is the number of variables. For sparse true DAGs, the number of operations of the PC algorithm is much less than $\mathcal{O}(2^N)$, which may not be the case with the MD-PC algorithm, depending on the dependencies between the variables. If $p$ is the maximum number of parents in the true DAG, the MD-PC algorithm must check CI tests on the maximum $p$ variables. Hence, the number of CI tests is bounded by $O(N^p)$ which is similar to the bound of the number of CI tests for the PC algorithm. So for sparse high-dimensional networks knowing the maximum number of parents helps to reduce the complexity of the MD-PC algorithm.

## 5 CONCLUSION

We addressed the problem of causality detection by focusing on identifying the "true" directed acyclic graph (DAG) of a causal model from the joint probability distribution of observed variables. Our investigation revealed that existing algorithms, such as PC and its modified versions, were unable to detect certain simple structures that violated the faithfulness assumption. To overcome this limitation, we introduced the concept of minimal dependence faithfulness, relaxing the faithfulness assumption to account for situations where variables exhibit dependencies that violate traditional faithfulness criteria. We defined minimal dependence as the set of variables on which a given variable depends, even though it may be independent of each individual member of that set. Our analysis revealed that minimal dependencies of size at least two violated faithfulness, prompting us to develop a modified version of the PC algorithm capable of detecting this type of unfaithfulness. This modified algorithm provided sound and complete results, allowing us to identify all possible candidates for the true DAG, even in cases where traditional algorithms failed. Additionally, we characterized the structure of minimal dependencies in the true DAG and showed that they formed v-stars or directed stars in certain configurations. These insights allowed us to propose a class dashed PDAG representation to simplify the representation of the MD-equivalence class. We also discussed the relationship between minimal dependence faithfulness and other minimality and faithfulness assumptions. Overall, our findings provide insights into the challenges of causality detection and offer practical solutions to improve the accuracy of causal model inference from observational data.

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

## A    APPENDIX

**Definition 9 (d-separation)** *(Ramsey et al., 2012)  Consider DAG $\mathcal{G}$ with node set $\mathcal{X}$. A trail $\mathcal{T}$ between two nodes $X, Y \in \mathcal{X}$ is* active *relative to (or given) a set of nodes $\mathcal{Z} \subseteq \mathcal{X}$ if* (i) *for each collider on $\mathcal{T}$, at least one of the descendants of the collider node is in $\mathcal{Z}$, and* (ii) *no other node on $\mathcal{T}$ is in $\mathcal{Z}$. The node subsets $\mathcal{X}_1, \mathcal{X}_2 \subseteq \mathcal{X}$ are* d-separated *given $\mathcal{Z}$, denoted $d-sep_{\mathcal{G}}(\mathcal{X}_1, \mathcal{X}_2 \mid \mathcal{Z})$, if there is no active trail between any node $X_1 \in \mathcal{X}_1$ and any node $X_2 \in \mathcal{X}_2$ given $\mathcal{Z}$. The set of all d-separations in $\mathcal{G}$ is denoted by $\mathcal{I}(\mathcal{G})$.*

**Example 4 (Non-Bernoulli distribution; non-deterministic)** *The following is a V-structure SCM where the collider node $X$ does not have a deterministic relationship with $Y_1$ and $Y_2$ and does not take a Bernouli distribution. The SCM consists of the variables $X, Y_1, Y_2$ and is defined by the equations*

$$\begin{cases} X := f(Y_1, Y_2, U) \\ Y_1 := \text{Bernouli}(1/2) \\ Y_2 := \text{Bernouli}(1/4) \end{cases}$$

*where $U \in \{1, 2, 3, 4, 5\}$ is the noise variable with the following distribution*

$$P(U) = \begin{cases} .2 & U = 1 \\ .1 & U = 2 \\ .1 & U = 3 \\ .1 & U = 4 \\ .5 & U = 5 \end{cases}$$

*and the function $f$ is defined as follows:*

$$f(0, 1, U) = \begin{cases} 0 & U = 1 \\ 1 & U > 1 \end{cases}, f(1, 0, U) = \begin{cases} 0 & U \le 2 \\ 1 & U > 2 \end{cases},$$

$$f(1, 1, U) = \begin{cases} 0 & U \le 3 \\ 1 & U > 3 \end{cases}, f(0, 0, U) = \begin{cases} 0 & U = 4 \\ 1 & U > 4 \end{cases}$$

*It can be verified that $\mathcal{Y} = \{Y_1, Y_2\}$ is the minimal dependence of $X$.*

**Example 5 (Non-Bernoulli distribution; non-deterministic)** *The following is another V-structure SCM with binary nodes $X, Y_1 \in \{1, 2\}$ and non-binary node $Y_2 \in \{1, 2, 3\}$:*

$$\begin{cases} X := f(Y_1, Y_2, U) \\ Y_1 := \text{Bernouli}(0.4) \\ Y_2 := \text{Multinomial}(0.6, 0.2, 0.2) \end{cases}$$

*where for brevity we only provide the conditional probability distribution of $X$:*

$$P(X = 0 \mid Y_1, Y_2) = \begin{cases} 0.65 & Y_1 = 0, Y_2 = 0 \\ 0.25 & Y_1 = 0, Y_2 = 1 \\ 0.1 & Y_1 = 0, Y_2 = 2 \\ 0.33 & Y_1 = 1, Y_2 = 0 \\ 0.6 & Y_1 = 1, Y_2 = 1 \\ 0.7 & Y_1 = 1, Y_2 = 0 \end{cases}$$

*Again, it can be verified that $\mathcal{Y} = \{Y_1, Y_2\}$ is the minimal dependence of $X$.*

**Example 6 (Water tank)** *Consider a water tank that has an input flow and an output flow (of equal debit) controlled by two valves. Let $X \in \{0, 1\}$ denote the water level of the tank, which is 0 if the water level equals some desired level, e.g., 1 meter above the base, and is 1 otherwise. Also, denote by $Y_1 \in \{0, 1\}$ and $Y_2 \in \{0, 1\}$ the valve statuses of the input and output flows, where a value of 0 means that the valve is open and 1 means that it is closed. Clearly, whether the water level changes from the desired level ($X$) is independent of just $Y_1$ (the inflow tap being open or closed) or just $Y_2$*

*(the output flow valve being open or closed), but is not independent of any of them once the other one is known. For example, given that the inflow valve is closed, then the output flow valve being open implies that the water level decreases, and being closed implies that the water level remains unchanged. It follows that $\mathcal{Y} = \{Y_1, Y_2\}$ is a minimal dependence for $X$.*

*Proof of Proposition 1.* We prove by contradiction. Assume that for some $X \in \mathcal{X}$, there exists a minimal dependence $\mathcal{Y}$ of $X$ where $|\mathcal{Y}| \geq 2$, and assume on the contrary that there exists some DAG $\mathcal{G}$ to which $P$ is faithful. In view of faithfulness, Condition (2) yields the absence of any active trail between $X$ and each node in $\mathcal{Y}$. We show that $X$ is d-separated from every node $Y_i \in \mathcal{Y}$ given $\mathcal{Y}_{-i}$. Otherwise, for some $i \in \{1, \ldots, n\}$, there must be an inactive trail between $X$ and $Y_i$ that becomes active upon the observation of $\mathcal{Y}_{-i}$. The only way this is possible is to have on the inactive trail including one or more colliders where each has a node in $\mathcal{Y}_{-i}$ as its descendant (or effect node). Denote by $V_1$ the closest node to $X$ on this trail that together with two other nodes $V_2$ and $V_3$ on this trail form the collider $V_1 \to V_2 \leftarrow V_3$ where some node $Y_k \in \mathcal{Y}_{-i}$ is a descendant of (or equals) $V_2$. Let $\mathcal{T}_1$ be the directed path from $X$ to $V_2$ on this trail and $\mathcal{T}_2$ be a directed path from $V_2$ to $Y_k$ Then the concatenated path $\mathcal{T}_1 \mathcal{T}_2$ is a collider-free and hence active trail from $X$ to a node in $\mathcal{Y}_{-i}$ and in turn $\mathcal{Y}$, a contradiction. Hence, for all $i \in \{1, \ldots, n\}$, $X \perp Y_i \mid \mathcal{Y}_{-i}$ belongs to $\mathcal{I}(\mathcal{G})$ but not to $\mathcal{I}(P)$ according to (1). This violates the I-map assumption, a contradiction. $\square$

*Proof of Proposition 2.* Eq. 6 follows from the third condition in Definition 1. To prove (5), let $\mathcal{Y}' = \{Y_1', \ldots, Y_k'\}, k < n$. Then in view of (6), $P(X \mid \mathcal{Y}') = P(X \mid \mathcal{Y}' \setminus \{Y_1'\})$. Next, by letting $\mathcal{Y}' \setminus \{Y_1'\}$ to be $\mathcal{Y}'$ in (6), and $Y$ to be $Y_2'$, we obtain $P(X \mid \mathcal{Y}' \setminus \{Y_1'\}) = P(X \mid \mathcal{Y}' \setminus \{Y_1', Y_2'\})$. Hence, by induction, it can be shown that $P(X \mid \mathcal{Y}') = P(X)$. To prove (3),let $\mathcal{Y} = \{Y_1, \ldots, Y_n\}$ and assume that (1) holds for $Y = Y_1$. We prove by contradiction. Assume on the contrary that (1) does not hold for some $Y \in \mathcal{Y}$, say $Y_2$, i.e., $X \perp Y_2 \mid \mathcal{Y} \setminus \{Y_2\}$. Then $P(X \mid \mathcal{Y}) = P(X \mid \mathcal{Y} \setminus \{Y_2\})$. Now, by letting $\mathcal{Y}' = \mathcal{Y} \setminus \{Y_2\}$, (5) results in $P(X \mid \mathcal{Y} \setminus \{Y_2\}) = P(X))$. Thus, $P(X \mid \mathcal{Y}) = P(X)$. On the other hand, again from (5) , $P(X \mid \mathcal{Y} \setminus \{Y_1\}) = P(X)$. Hence, $P(X \mid \mathcal{Y}) = P(X \mid \mathcal{Y} \setminus \{Y_1\})$, yielding $X \perp Y_1 \mid \mathcal{Y} \setminus \{Y_1\}$, a contradiction, completing the proof. Finally, (4) is also proven by contradiction as if on the contrary $X \perp \mathcal{Y}$, then $P(X \mid \mathcal{Y}) = P(X)$ which similar to the proof for (5) results in a contradiction. $\square$

*Proof of Theorem 1.* Consider an arbitrary $Y_i \in \mathcal{Y}$. Should $Y_i$ be adjacent to $Z$ the result is trivial, so consider otherwise. In view of Condition (1), $Z \not\perp Y_i \mid \mathcal{Y}_{-i}$. So $\mathcal{Y}_{-i}$ observed, $\mathcal{G}$ being an I-map implies an active trail, say $\mathcal{T}$, between $Z$ and $Y_i$. Thus, every collider on trail $\mathcal{T}$ becomes active upon observing $\mathcal{Y}_{-i}$, implying that the effect node or its descendant is in $\mathcal{Y}_{-i}$, and hence, $\mathcal{Y}$. $\square$

*Proof of Theorem 2.* Consider an arbitrary dependent member $Y \in \mathbf{Dep}^{\mathrm{o}}(Z)$. Should $Y$ be adjacent to $Z$ the result is trivial, so consider otherwise. Let $\mathcal{Y} \in \mathbf{Dep}(Z)$ be the minimal dependence set that includes $Y$, i.e., $Y \in \mathcal{Y}^{\mathrm{o}}$. In view of (7), Once $\mathcal{Y} \setminus \{Y\}$ is observed, $\mathcal{G}$ being an I-map implies an active trail between $Z$ and $Y$ that does not become inactive whether a node other than those in $\mathcal{Y}$ is observed. Hence, $Z$ and $Y$ must form a collider with the effect node or its descendant in $\mathcal{Y} \setminus \{Y\}$. On the other hand, Assumption 3 excludes links between $Z$ and non-dependent members. Hence, the effect node of the collider must belong to $\mathbf{Dep}^{\mathrm{o}}(Z)$. Thus, $Z$ and $\mathbf{Dep}^{\mathrm{o}}(Z)$ admit a v-star centered at $Z$. $\square$

*Proof of Corollary 1.* The proof follows Part *(ii)* in Theorem 2 and the fact that if a minimal dependence $\mathcal{Y} \in \mathbf{Dep}(Z)$ is a singleton, say $\{Y\}$, then $\mathcal{Y} \setminus \{Y\}$ becomes empty, implying that a collider cannot connect $Y$ and $Z$. So the two are adjacent. $\square$

*Proof of Corollary 2.* Let $\mathcal{Z} = \{Z_1, \ldots, Z_n\}$ be a symmetric minimal dependence. Theorem 2 implies that either *(i)* $Z_i$ and $Z_j$ are linked, *(ii)* $Z_i$ and $Z_j$ form a collider with an effect node $Z_{ij} \in \mathcal{Z}_{-ij}$, or *(iii)* $Z_i$ and $Z_j$ form a collider with an affect node $V_{ij} \in \mathcal{X} \setminus \mathcal{Z}$ with some node $Z_{ij} \in \mathcal{Z}_{-ij}$ being a descendant of $V_{ij}$. Let $\mathcal{T}$ be the longest directed path among the nodes in $\mathcal{Z}$ and let $Z$ be the last node in this path. Node $Z$ cannot be linked to any of the nodes in $\mathcal{T}$ as that would result in a directed cycle. Also, node $Z$ cannot have an outgoing link to any of the other symmetric minimal dependence nodes $\mathcal{Z}$ as that would result in a longer directed path. Hence, if we denote by $\mathcal{S} = \{S_1, \ldots, S_k\}$ those nodes in $\mathcal{Z}$ that are not linked to $Z$, it follows from the three aforementioned cases that each $S_i$ forms a collider with $Z$ with some effect node $V_i \in \mathcal{X} \setminus \mathcal{Z}$. The descendant of $V_i$ cannot be any of the nodes that are linked to $Z$ as then that node together with $V_i$ and $Z$ form a directed cycle. Hence, each $V_i$ takes as its descendant a node in $\mathcal{S}$, which we denote by $S_{i+1}$. However, since $\mathcal{S}$ is finite, it holds for some $i$ that $S_{i+1} = S_j$ for some $j < i$.

Then $S_j \to V_j - \to S_{j+1} \to V_{j+1} \ldots V_i - \to S_j$ forms a directed cycle which is impossible ($- \to$ denotes a directed path). Thus, $\mathcal{S}$ is empty, implying that every node in $\mathcal{Z} \setminus \{Z\}$ is connected to $Z$. $\square$

*Proof of Theorem 3.* Denote by $\mathcal{G}'$ the class dashed PDAG of $P$. (soundness) Consider adjacent nodes $X$ and $Y$ in the true DAG $\mathcal{G}$. In view of Assumption 3, $Y$ is a dependent member of some minimal dependence $\mathcal{Y} \in \mathbf{Dep}(X)$, i.e., $Y \in \mathcal{Y}^{\mathrm{o}}$. *Case 1.* Should $|\mathcal{Y}| = 1$, no set $\mathcal{U} \subseteq \mathcal{X} \setminus \{X, Y\}$ renders $X$ and $Y$ conditionally independent, and hence, the connecting edge would not be eliminated neither by the PC part of the algorithm, nor by the beginning part when it finds minimal dependencies of size at least 2. Hence, the edge between $X$ and $Y$ exists in the output of the algorithm and will not be dashed. So one of the following cases holds. *Case 1.1.* The edge is directed, say $X \to Y$ in $\mathcal{G}'$. Then in view of Assumption 4, the direction is either because of the existence of another symmetric minimal dependence $\mathcal{Z} \cup \{X\}$ for some $\mathcal{Z} \in \mathbf{Dep}(X)$, where no node in $\mathcal{Z}$ is adjacent with $Y$, or that the opposite direction would cause a directed cycle or a new immorality. In either case, the algorithm correctly identifies the direction. *Case 1.2.* The edge is undirected in $\mathcal{G}'$. Then again in view of Assumption 4, none of the possibilities in Case 1.1 take place which is the same as with the algorithm. *Case 2.* Should $|\mathcal{Y}| \geq 2$, the edge would be dashed in $\mathcal{G}'$ (except for the already covered case when it is directed due to Assumption 4), which is correctly identified by the algorithm, because it exhaustively searches through all subsets of variables that are marginally independent of $X$, identifies $\mathcal{Y}$ as a minimal dependence of $X$, and in turn identifies $\mathcal{Y}^{\mathrm{o}}$ and marks the links from $X$ to $\mathcal{Y}^{\mathrm{o}}$ as dashed. (completeness) Consider nodes $X$ and $Y$ that are not adjacent in the true DAG $\mathcal{G}$. Then $X$ and $Y$ are d-separated in $\mathcal{G}$, i.e., $(X \perp Y \mid \mathcal{V}) \in \mathcal{I}(\mathcal{G})$ for some $\mathcal{V} \in \mathcal{X} \setminus \{X, Y\}$. As $\mathcal{G}$ is an I-map for $P$, $(X \perp Y \mid \mathcal{V}) \in \mathcal{I}(P)$. Should $X \perp Y \notin \mathcal{I}(P)$, then $Y$ does not belong to a minimal dependence of $X$ and the link between them is eliminated by the algorithm as it would have been in normal PC. The same holds if $X \perp Y \in \mathcal{I}(P)$ but there is no superset $\mathcal{Y} \supset Y$ for which $(X \perp \mathcal{Y}) \notin \mathcal{I}(P)$ and $X$ is independent of each member of $\mathcal{V}'$. The only case left is when $X \perp Y \in \mathcal{I}(P)$ and there is a superset $\mathcal{Y} \supset Y$ that is a minimal dependence of $X$, but then all the connections between $X$ and dependent members $Y \in \mathcal{Y}^{\mathrm{o}}$ are a dashed edge in $\mathcal{G}'$ which is also the case with the output of the algorithm and there is no edge between $X$ and the non-dependent members $Y \in \mathcal{Y} \setminus \mathcal{Y}^{\mathrm{o}}$ in view of Assumption 3, which is again the case with the algorithm. $\square$

**Example 7** *Consider the SCM with variables $\mathcal{X} = \{X, Y_1, Y_2, Y_3, Y_4, Y_5, Y_6\}$ defined by*

$$Y_i := \mathrm{Bernoulli}(1/2), \quad i = 1, 2, 3, 4, 5$$
$$X := Y_1 \wedge (Y_2 \oplus Y_3 \oplus Y_4),$$
$$Y_6 := X \oplus Y_5,$$

*where $\wedge$ is the logical conjunction. Fig.3-a) shows the true causal DAG and Fig.3-b) shows the resulting network from the PC algorithm, which does not capture the links between $X$ and $Y_2, Y_3, Y_4$ and between $Y_6$ and $X$ and $Y_5$. The minimal dependence sets of each variable are as follows:*

$$\mathbf{Dep}(X) = \{\{Y_1\}, \{Y_2, Y_3, Y_4\}, \{Y_5, Y_6\}\},$$
$$\mathbf{Dep}(Y_1) = \{\{X\}\}, \quad \mathbf{Dep}(Y_2) = \{\{X, Y_3, Y_4\}\},$$
$$\mathbf{Dep}(Y_3) = \{\{X, Y_2, Y_4\}\}, \quad \mathbf{Dep}(Y_4) = \{\{X, Y_2, Y_3\}\},$$
$$\mathbf{Dep}(Y_5) = \{\{X, Y_6\}\}, \quad \mathbf{Dep}(Y_6) = \{\{X, Y_5\}\}$$

*Consequently, the symmetric minimal dependence is*

$$\mathbf{Sym} = \{\{X, Y_1\}, \{X, Y_2, Y_3, Y_4\}, \{X, Y_5, Y_6\}\}.$$

*So following the MD-PC algorithm, we first obtain the graph in Figure 3-b). The solid edge between $X$ and $Y_1$ is because $\{X\}$ is the only member of $\mathbf{Dep}(Y_1)$. Both $\{X, Y_1\}$ and $\{X, Y_2, Y_3, Y_4\}$ are symmetric minimal dependencies. Moreover, there is no link between $Y_1$ and $\{X, Y_2, Y_3, Y_4\}$ (none are in the dependence set of the other), and $Y_1 \not\perp Y_2, Y_3, Y_4 \mid X, \mathcal{U}$ for all $\mathcal{U} \subseteq \{Y_5, Y_6\}$, implying that $X \notin \cup_{Y' \in \{Y_2, Y_3, Y_4\}} \mathbf{Sepset}(Y_1, Y')$, it follows that $Y_1, Y_2, Y_3, Y_4$ and $X$ form a star directed at $X$, resulting in Figure 3-c). This is not the case for symmetric minimal dependencies $\{X, Y_1\}$ and $\{X, Y_5, Y_6\}$ for example.*

*To avoid a new immorality between $Y_1$ and $Y_5$ and between $Y_1$ and $Y_6$, the edges $XY_5$ and $XY_6$ may not be pointing to $X$. In view of the fact that $Y_5, Y_6$, and $X$ form a directed star, this implies that the directed star is either centered at $Y_6$ or $Y_5$. These results are in the graphs in Fig.4.*

---

**Algorithm 1:** The MD-PC Algorithm

---

**Input:** A set of random variables $\mathcal{X}$ and their joint probability distribution $P$
**Output:** A set of partially directed acyclic graphs with nodes $\mathcal{X}$

1 Form the complete undirected graph $\mathcal{G}$ over nodes $\mathcal{X}$;
2 $\mathbf{Sym} = \emptyset$;                    // The set of symmetric minimal dependencies
3 $\mathbf{Perp}(X) = \emptyset$ for all $X \in \mathcal{X}$;    // The set of variables independent of $X$
4 $\overline{\mathbf{Dep}}(X) = \mathbf{Dep}^{\mathrm{o}}(X) = \emptyset$ for all $X \in \mathcal{X}$;
5 $\mathbf{Sepset}(X, Y) = \emptyset$ for all $X, Y \in \mathcal{X}$;
6 **for** $X \in \mathcal{X}$          // find minimal dependencies of size at least 2
7    **for** $Y \in \mathcal{X} \setminus \{X\}$
8       **if** $X \perp Y$
9          $\mathbf{Perp}(X) \leftarrow \mathbf{Perp}(X) \cup \{Y\}$;
10    **for** $Y \in \mathbf{Perp}(X)$
11       **for** $\mathcal{V} \subseteq \mathbf{Perp}(X) \setminus \{Y\}$          // Sort $\mathcal{V}$ from small to large
12          **if** $\nexists \mathcal{Y} \in \mathbf{Dep}(X) : \mathcal{V} \cup \{Y\} \subseteq \mathcal{Y}$          // Due to Condition 3 in Definition 2
13             **if** $X \not\perp Y \mid \mathcal{V} \cup \mathcal{U}$ *for all* $\mathcal{U} \subseteq \mathcal{X} \setminus (\{X, Y\} \cup \mathcal{V})$
14                $\mathbf{Dep}(X) \leftarrow \mathbf{Dep}(X) \cup \{\{Y\} \cup \mathcal{V}\}$;
15                $\overline{\mathbf{Dep}}(X) \leftarrow \overline{\mathbf{Dep}}(X) \cup \{Y\} \cup \mathcal{V}$;
16                $\mathbf{Dep}^{\mathrm{o}}(X) \leftarrow \mathbf{Dep}^{\mathrm{o}}(X) \cup \{Y\}$;
17                **for** $V \in \mathcal{V}$          // Find other dependent members
18                   **if** $X \not\perp V \mid \{Y\} \cup (\mathcal{V} \setminus \{V\}) \cup \mathcal{U}$ *for all* $\mathcal{U} \subseteq \mathcal{X} \setminus (\{X, Y\} \cup \mathcal{V})$
19                      $\mathbf{Dep}^{\mathrm{o}}(X) \leftarrow \mathbf{Dep}^{\mathrm{o}}(X) \cup \{V\}$;
20    Remove the edges between $X$ and $\overline{\mathbf{Dep}}(X)$ in $\mathcal{G}$;
21 **for** $X \in \mathcal{X}$                    // Add symmetric minimal dependencies
22    **for** $\mathcal{Y} \in \mathbf{Dep}(X)$
23       **if** $\mathcal{Y} \setminus \{Y\} \in \mathbf{Dep}(Y)$ *for all* $Y \in \mathcal{Y}$
24          $\mathbf{Sym} \leftarrow \mathbf{Sym} \cup \{\mathcal{Y} \cup \{X\}\}$;
25 $m = 0$
26 **while** *maximum node degree in* $\mathcal{G}$ *is greater than* $m$ **do**          // Normal PC
27    **for** $X \in \mathcal{X}$
28       **for** $Y \in \mathrm{Adj}(\mathcal{G}, X)$
29          **for** $\mathcal{U} \subseteq \mathrm{Adj}(\mathcal{G}, X) \setminus \{Y\}$ *and* $\mid \mathcal{U} \mid = m$
30             **if** $X \perp Y \mid \mathcal{U}$
31                Remove the edge $X - Y$ from $\mathcal{G}$;
32                $\mathbf{Sepset}(X, Y) \leftarrow \mathcal{U}$;
33    $m = m + 1$;
34 **for** $X \in \mathcal{X}$   // Remaining edges form a singleton minimal dependence
35    **for** $Y \in \mathrm{Adj}(\mathcal{G}, X)$
36       $\mathbf{Dep}(X) \leftarrow \mathbf{Dep}(X) \cup \{\{Y\}\}$;
37       $\mathbf{Sym} \leftarrow \mathbf{Sym} \cup \{\{X, Y\}\}$;
38 **for** $X \in \mathcal{X}$
39    **for** $\mathcal{Y}_1, \mathcal{Y}_2 \in \mathbf{Dep}(X)$                    // Orientation
40       **if** $\mathcal{Y}_1 \cup \{X\}, \mathcal{Y}_2 \cup \{X\} \in \mathbf{Sym}$
41          **if** *there is no edge between a node in* $\mathcal{Y}_1$ *and a node in* $\mathcal{Y}_2$
42             **if** $X \notin \cup_{Y_1 \in \mathcal{Y}_1, Y_2 \in \mathcal{Y}_2} \mathbf{Sepset}(Y_1, Y_2)$
43                Link every node in $\mathcal{Y}_1 \cup \mathcal{Y}_2$ to $X$ with a directed edge;
44    **for** $\mathcal{Y} \in \mathbf{Dep}(X)$                    // Dashed edges
45       **if** $|\mathcal{Y}| \geq 2$
46          Connect $X$ and each node in $\mathcal{Y} \cap \mathbf{Dep}^{\mathrm{o}}(X)$ by dashed edges, unless they are already connected by a directed or undirected non-dashed edge;
47 Orient every undirected edge if the opposite direction yields a directed cycle or a new immorality;

---

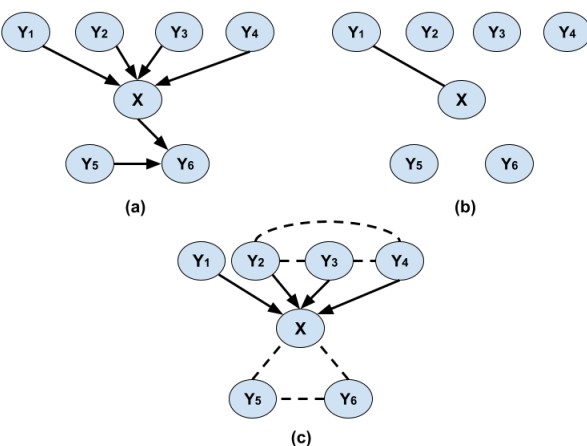

Figure 3: **a)** The true DAG for SCM in Example 7. **b)** Output of the PC algorithm **c)** The dashed PDAG representing the MD-equivalent class from the MD-PC algorithm.

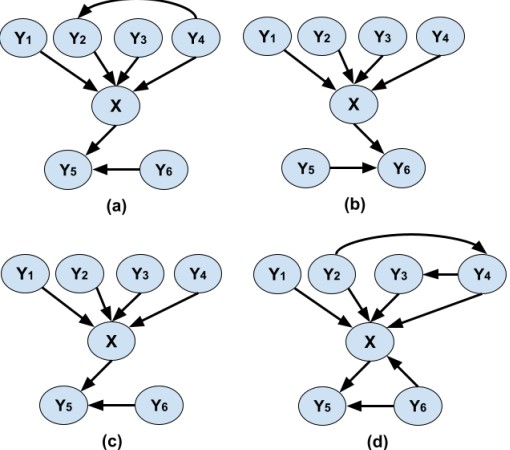

Figure 4: Examples of the MD-equivalent class.