# OpenReview forum: "Structure Learning for Unfaithful Distributions: The Minimal Dependence Faithfulness"
_ICLR.cc/2025/Conference — Submitted to ICLR 2025_

### Official Review · Reviewer_VrEs · 2024-10-31

**Soundness:** 2
**Presentation:** 1
**Contribution:** 2
**Rating:** 3
**Confidence:** 4

**Summary:**

The paper addresses the question of identifying a causal DAG purely though observational data. Traditional approaches require faithfulness, i.e., that the distribution does not exhibit "extra" independencies that are not implied by the underlying structure.  The authors define notions of "weak/strong minimal dependence", and prove various properties about them.  The authors use these properties to show that, under different assumptions (relating to the these minimal dependences), a certain algorithm can recover the set of PDAGs that could have given rise to the distribution.

**Strengths:**

The authors identify a real problem with previous approaches and perform a deep and complex technical investigation into these failure modes. They reframe the problem in a novel way (although I don't see its utility), and prove a number of seemingly non-trivial results about their definitions.

**Weaknesses:**

The paper has many weaknesses.

**1.** First and most glaringly, the presentation (especially the first half) is full of mathematical mistakes, and imprecise wording that is false.  I believe there are a large number of typos in critical sections, backwards definitions, and minor misunderstandings of the literature. I've detailed many of these at the end of the response.

**2.** Secondly, and perhaps ultimately most importantly, I do not see the utility of the final result.  The main result of the paper seems to be Theorem 3, which claims to show that their main algorithm (MD-PC / Algorithm 1, which, by the way, was suspiciously deferred to the appendix) solves the desired problem under certain assumptions.  Yet these new assumptions are far more complex and difficult to understand than the old ones, and I have no reason to believe that they are plausible.  They are built on a tower of complicated math, whose implications I cannot imagine fully understanding without spending another week working them out.

The end result of the algorithm is a class of PDAGs that contains the "true" causal DAG G. But how long does it take to generate this? It is mentioned that it takes exponential time, and unlike prior work, they have no reason to believe that this will be smaller if the true DAG is sparse, if the independencies are bad enough.  Now, this may be unavoidable --- but take an outside view.   With an exponential number of conditional independence tests, you can simply output I(G), which is even better and more precise than the PDAG class up-to-MD-Equivalence that they guarantee.

**3.** There is no experimental evaluation, nor any attempt to implement the algorithm itself.  Given the imprecision in the math at the beginning, the lack of experimental evaluation is concerning.  ICLR is a conference that, perhaps moreso than other ML conferences, prides itself on rigorous evaluation, bringing me to the final point: even if these (substantial) concerns were addressed, I have my doubts as to whether this would be a good fit for ICLR.  There isn't much "learning" involved; the representation itself is computed directly in one pass, and the results are purely theoretical.

Ultimately, while there are some interesting ideas here, and it clearly represents a lot of mathematical thought, it needs serious reworking and simplification to be of use to the general community. And it needs significant empirical backing to be of use to this community in particular.

I left line-by-line comments for the first six pages, before skipping to the main results and trying to understand the contribution at a high level.  Unfortunately, while the investigation is thorough, it does not seem very meaningful or impactful, and it is riddled with small technical mistakes and holes that make it difficult to trust. Here are my inline comments:

----------

ABSTRACT.
* 013: PC undefined.
* 014: The word "assumption" in "restrictive faithfulness assumption" is more of a guarantee, as it is a property of the DAG generated, and not an assumption about the input distribution.

INTRO
* 039: I feel a disclaimer is necessary: it is well-known that the distribution of the observed variables has nowhere near enough information to describe the effects of interventions.  What has changed in recent decades to make this seem more palatable or interesting?
* 042: "Imply" is the wrong word; "means" is closer.
* 043: I think a simple example would help: two independent variables X, Y, with the dag  X -> Y.  Especially because this is the focus of the paper, and you want to keep in mind the meaning of "faithfulness". Currently, it's not even defined directly; the focus is on the converse  (not the "inverse", by the way).
* 048: missing word ("partially DAG" doesn't parse)
* 049: I don't believe that there is a unique PDAG with some property that represents an arbitrary Markov equivalence class, and so I object that there is a "true PDAG" of the joint probability distribution.
* 053: expand on the "practical situations" in which faithfulness fails; after all, that is your motivating case of interest!
* 059: Unhelpful sentence structure, which suggests {adjacency/orientation}-faithfulness were already defined, when in fact they are defined in this sentence.
* 060: style violation; reference should be text style, parens only around year.
* 060: We still never hear what PC stands for.
* 071: missing word ("smallest"?)
* 071-074: misplaced definitions, at a weird intermediate level of formality. Either describe them more intuitively here, along with their role in the paper, or add enough formality and make it correct (ideally later on).
* 073*: why should there be a unique (smallest?) such set?  Edit: I see that there is not; the wording must be clarified.

PROBLEM FORMULATION
* 081: I believe Peters is the wrong reference for this definition (it should be Pearl 2009 / SGS 2000).
* 081: It's just a variable; you're not using the concept of a random variable. Also, if \X are "the random variables", then this suggests it should contain \N, which are also claimed to be "random variables".
* 088**: claim (i) is false, unless you also assume that G is a DAG.  Claim (ii) is also false because you did not assume that the relationship between the functions \F is acyclic. I also believe it is misrepresented as a claim; in fact, it is essentially directly part of the data of a SCM if you write down the definition carefully.
* 094: the term "observational distribution" is not defined; it should be clarified that this is a distribution over \X.
* 098: missing colon
* 100: missing [The] true DAG.
* 100: give reference for the independencies of a DAG.
* 115-117: this is not really a "different perspective"; it's well-known to be equivalent.  Furthermore, the thread of the story comes apart: this assumption does not fix "the motivating example" in lines 096-099. So why bother formulating problem 2, and why tell the story in this order?
* 117*: This example only makes things worse.  Why are we talking about the empty graph as a potential solution to problem 2?  Given a typical distribution P that does not come from an SCM in which every variable is independent, the empty graph is clearly just the wrong answer.

* 136**: I don't buy that I-equivalent DAGs can be represented by a PDAG (at least as I understand the concept; it is not defined here). Please give a more precise reference to a specific reference in Koller & Friedman (2009).

* 146: ":=" is the wrong symbol; Y is a variable, not a distribution. You mean "~". Also, you have to say that these are independent for the example to work.  Note also you're sweeping the noise variables under the rug. It might be easier (as well as more precise) to use them.
* 149: I've never heard of "probabilistic arithmetic" before.

* 170: I don't get it --- are Figure 1(a) and 1(b) not identical? Does this not contradict what was seen earlier?


MINIMAL DEPENDENCE FAITHFULNESS

* 195*: Why do this with just a single variable Y? Why not a subset? That seems more natural to me.
* 210: I disagree that this is counter-intuitive (with a dash).
* 224: Rephase: "knowing any *one* of these symptoms does not make *either* disease more likely".  Any/Any is very ambiguous.
* 220: Example 3 is good, but the parentheticals are clunky.
* 225: A quantifier is used; it's just an existential one. You are arguing it's equivalent to the universal one.
* 262: The language suggests there is a unique weak minimal dependence, which is not the case.

* I don't see why Theorem 1 says anything useful for determining the structure. It guarantees that certain edges must exist, which places upper bounds on the strength of the independences of the output DAG. How, for example, does this keep us from outputting the trivial DAG  in which the parents of Xi are all Xj for j < i?

* 273: I assume based on context and typeface, that \U should be a set of variables ( a subset of \X \setminus {X} \cup \Y).  This would also explain the terminology "strong".

* 311: Based on usage below, you should not be using \X for this set.  Also, there are many such DAGs, so "the DAG" is incorrect.
* 316: "centered at Z" not defined.
* 370: P-maps are introduced here; sentence structure again is not introducing but recalling.
* 379: This is confusing, because P-maps were not defined. What does it mean for Y1 and Y2 to be "adjacent" in a P-map (not a PDAG or DAG)?

* 500: Assumptions 3 and 5 are not just about G, but also the SCM, and P as well.

PROOFS
* 666-667:  I don't think (6) alone follows from condition 3 of definition 1. Why should it be the first and not the second condition that fails for this proper subset?

**Questions:**

* why should we have any confidence that assumptions 3 and 4 are satisfied?
* What makes Algorithm 1 better than brute force search over graphs?
* Does the result degrade naturally for approximate CI tests?

I left more specific comments in the line-by-line comments of the previous section.

---

### Official Review · Reviewer_h3Ti · 2024-11-01

**Soundness:** 3
**Presentation:** 2
**Contribution:** 3
**Rating:** 5
**Confidence:** 4

**Summary:**

The authors proposed to relax the faithfulness and assume a weaker version known as minimal dependence faithfulness for causal discovery from observational data. The authors argue that there is no existing algorithm that is capable of learning the causal structure when there is a variable resulting from the XOR relation of several Bernoulli variables. The authors show that this type of unfaithfulness can generally appear in examples where a single variable is independent of other variables individually, but is dependent on one of them once it’s conditioned on the remainder of the set. The paper also gives a weaker version of orientation faithfulness and provides a sound and complete algorithm based on a modification of the PC algorithm under those weaker assumptions. Overall, I think the manuscript’s writing should be improved. One issue I have is that there is a lack of discussion on how strong the minimal dependence faithfulness assumption is compared to the sparest Markov representation (SMR) or frugality. It also does not include any experiments to demonstrate the algorithm's benefits over those that are developed based on SMR, which I think is important to at least show the importance of such a weak version of faithfulness.

**Strengths:**

- The paper relaxes the faithfulness assumption and proposes a modification of the PC algorithm based on a weaker assumption for a very special setup under XOR relations.
- The problem setup is motivated by practical scenarios (See example 3).
- The proofs seem to be correct.
- The paper provides examples on distributions that exist for the problem.
- The comparison with closely related work is provided in section 3.4.
- The complexity of the proposed algorithm is provided.

**Weaknesses:**

- The organization of the introduction and related work should be improved. I will have read until the end of the paper in order to gain more clarity on the introduction. I summarize my recommendations below.

      - First of all, it is not clear why one should care about the causal structure from a variable resulting from XOR of several Bernoulli variables specifically. It will be helpful to bring up Example 3 in the introduction.

      - There are many algorithms based on assumptions that are weaker than faithfulness that are not even cited in the paper. For example, to name a few [1,2,3,7, 9]. The introduction does not mention any advantages of using constraint-based methods with the assumption of faithfulness over others. For example, LiNGAM does not rely on the assumption of faithfulness at all [8].  Also, more classic literature talks about how faithfulness can be violated [4,5].

      - Also, the paper cited [6] for this paragraph "Under the causal Markov and faithfulness assumptions, constraint-based approaches have been shown to correctly find the true PDAG given the joint probability distribution, and the same holds asymptotically with respect to the number of data instances of a given dataset”,  This is misleading because [6] actually relies on an assumption that is strictly is weaker than faithfulness.

      - Lines 63-65: "all of these and other existing algorithms, such as MGM-FCI-MAX (Horii, 2021), fail to detect the true graph in examples such as the generalized XOR where several Bernoulli random variables cause another variable.” this is inconsistent with the paragraph mentioned in line 159 in section 2 where it talks about there is an exception due to the work by Marx et al. 2021.

- While the authors provide some practical scenarios for the problem setup, there is no real-world experiment nor synthetic experiment to substantiate the results, which makes me not sure how much of a difference the proposed algorithm can make in practice.
- According to Marx et al 2021, “tripe unfaithfulness” in line 459 should be “unfaithful triples”.

References:
- [1] Wienöbst, Marcel, and Maciej Liskiewicz. "Recovering causal structures from low-order conditional independencies." Proceedings of the AAAI Conference on Artificial Intelligence. Vol. 34. No. 06. 2020.
- [2] Kocaoglu, Murat. "Characterization and learning of causal graphs with small conditioning sets." Proceedings of the 37th International Conference on Neural Information Processing Systems. 2023.
- [3] Lee, Kenneth, Bruno Ribeiro, and Murat Kocaoglu. "Constraint-based Causal Discovery from a Collection of Conditioning Sets." 9th Causal Inference Workshop at UAI 2024. 2024.
- [4] Uhler, Caroline, et al. "Geometry of the faithfulness assumption in causal inference." The Annals of Statistics (2013): 436-463.
- [5] Robins, James M., et al. "Uniform consistency in causal inference." Biometrika 90.3 (2003): 491-515.
- [6]Ng, Ignavier, et al. "Reliable causal discovery with improved exact search and weaker assumptions." Advances in Neural Information Processing Systems 34 (2021): 20308-20320.
- [7] Lam, Wai-Yin, Bryan Andrews, and Joseph Ramsey. "Greedy relaxations of the sparsest permutation algorithm." Uncertainty in Artificial Intelligence. PMLR, 2022
- [8] Shimizu, Shohei, et al. "A linear non-Gaussian acyclic model for causal discovery." Journal of Machine Learning Research 7.10 (2006).
- [9] Sadeghi, Kayvan. "Faithfulness of probability distributions and graphs." Journal of Machine Learning Research 18.148 (2017): 1-29.

**Questions:**

- Why don't the authors at least include some synthetic experiments based on various distributions described in the paper such as Examples 4,5, and 7 to show the utility of the proposed algorithm over PC and those that are developed based on Sparest Markov Representation such as GRaSP [1]?

If the authors can include some experimental results to show the value of the algorithm, I will raise my score.

[1] Lam, W. Y., Andrews, B., & Ramsey, J. (2022, February). Greedy Relaxations of the Sparsest Permutation Algorithm. In The 38th Conference on Uncertainty in Artificial Intelligence.

---

### Official Review · Reviewer_DvQE · 2024-11-02

**Soundness:** 2
**Presentation:** 1
**Contribution:** 3
**Rating:** 3
**Confidence:** 3

**Summary:**

The paper considers the problem of causal discovery under a weaker faithfulness assumption exemplified by XOR gates with Bernoulli inputs. It proposes the notion of minimal dependence (MD) to capture such parameter-specific faithfulness and shows how it is reflected on causal graphs. It also introduces the notion of dashed PDAG to classify the MD-equivalence class of the given distribution. Algorithm is developed to discover such dashed PDAGs.

**Strengths:**

1. According to my knowledge, the contribution made by this paper is novel.
2. The work contains theoretical results and the proofs are included in the appendix. The full algorithm is also provided in the appendix.

**Weaknesses:**

1. I think the main weakness is its presentation. I found it difficult to understand the texts. For example, I still don't understand the last paragraph of introduction.
2. The paper proposed two different versions of minimal dependence: weak minimal dependence and strong minimal dependence. However, unless I missed something, I don't see how the weak minimal dependence is used anywhere for the causal discovery. I'm not sure if this definition and its propositions are needed.
3. I think a main theorem that is missing from the paper is that all DAGs in an MD-equivalent class admit the I-map of the given distribution. Without this theorem, readers cannot appreciate the notions of MD-equivalence and dashed PDAG.
4. More details should be added to the discussion of the MD-PC algorithm. For example, the paper should mention which specific theorems/corollaries are used by the algorithm.

Minor points:
  - Figure 1(a) and 1(b) are completely identical. Not sure what the difference is.
  - Incorrect labels of $Y_{n-1}$ and $Y_{n}$ in Figure 1(c)
  - Section 3 "Should faithfulness be in force, we would conclude that X and Y are connected by some active trail." Not sure about the meaning here.
  - Based on the context, I think it should be $U \subseteq X \setminus (\{X\} \cup Y)$ in Definition 2 (7)? Otherwise, I don't think strong minimal dependence implies the weak independence.
 - It's better to spell out the entire "weak minimal dependence set" to avoid confusion with "(strong) minimal dependence".

**Questions:**

1. Paragraph below Proposition 2: "This means that in practice, if for a ...." I don't understand the purpose of the text. Are you trying to provide a method for finding a minimal dependence set? If so, I don't think it is explicit enough.
2. In Theorem 2 "..., where $\mathcal{Y}$ is the minimal dependence of Z that includes Y", does it mean for some $\mathcal{Y}$ or for all $\mathcal{Y}$? Is this minimal dependence $\mathcal{Y}$ unique? Also, doesn't statement (i) imply  statement (ii)?
3. Remark 1 after Theorem 3 (btw, it should be Remark 2), why is the number of CI tests required by the MD-PC algorithm exponential in maximum number of parents rather than the maximum degree?
4. How complete/informative is the learned dashed PDAG? That is, how likely that the dashed PDAG returned by MD-PC is the minimal one? Examples will be helpful.

---

### Official Review · Reviewer_2fnM · 2024-11-04

**Soundness:** 4
**Presentation:** 2
**Contribution:** 2
**Rating:** 5
**Confidence:** 4

**Summary:**

This paper investigates into relaxing the faithfulness assumption in causal discovery and proposes a relaxed faithfulness, called minimal dependence faithfulness. Minimal dependence of a variable is a set of variables which individually independent to the variable but becomes dependent to the variable given the rest of the set. After introducing further restricted classes of minimal dependence (strong, symmetric), the paper presents the equivalence class of a DAG under minimal dependence (with additional assumptions to further orient edges), and devises a modified PC algorithm to detect minimal dependence and orient edges involving minimal dependencies.

**Strengths:**

- The problem is well motivated: relaxing faithfulness is one of the most important topics in causal discovery.
- The concept of minimal dependence is easy to follow. Its relationships to previous notion of (weak) types of faithfulness are well explained.
- The paper not only investigate into the implied properties of minimal dependence, but also covers equivalence class and a (theoretical) causal discovery algorithm.

**Weaknesses:**

- This paper employs causal sufficiency which is not explict as “assumption” but hidden in the definition of SCM in line 083.
- How much can we expect that a given distribution exhibits ‘minimal dependence’ (isn’t it Lesbegue measure zero?) BTW, I am very thankful for the authors for coming up with examples other than XORs.
- Despite of MD-PC’s theoretical theoretical guarantee, causal discovery almost always runs on a finite sample. It is crucial to empirically investigate into how MD-PC actually detects minimal dependencies. In some cases where no minimal dependence exists, MD-PC’s performance might (certainly?) be worse than traditional PC. (I “hopefully” expect that the false positive rate of minimal dependence is very low given that the sample size is often small or CI test is not so powerful. The issue will be true positive rate, unless we artificially create strong conditional dependence for minimal dependencies through carefully constructed synthetic datasets.)
- The paper may require more visual elements although visualizing some of them may not be very obvious (e.g., Dep^o is more about functional/probabilistic aspects than (purely) graphical porperty.)
- The organization of the paper makes readers somewhat exhausted with definitions and assumptions throughout the paper. One issue is that the authors present weak, strong, and then symmetric version where the the weak version is later dropped (i.e., unused). Further, MD-PC and equivalence only applies to the strong and symmetric version of minimal dependence. While it is the authors’ choice to build up notations slowly, sometimes it would be better to present what the authors feel “matter”.

**Questions:**

- Why is the minimal dependence, “minimal”? Is this because of “no proper subset…” part or something else with respect to faithfulness/dependence itself not about the set.
- Can we have examples for Definition 2 showing the difference between Y^o and Y clearly?
- Def. 2 U “\in” should be “\subseteq”
- Line 251 uniform → universal?
- Line 460 tripe → triple
- Consider replacing ‘links’ to ‘edges’
- Citation for FCI algorithm should be of Spirtes not a survey paper. (Line 046)

Regarding "Contributions", there is a question about "Are the results valuable to share with the *broader* ICLR community?" I guess this paper would be a perfect fit for UAI or AAAI.

**Details Of Ethics Concerns:**

.

---

> ### Comment · Reviewer_2fnM · 2024-11-26
>
> No rebuttals. Other reviewers share the similar concerns. Hence, I will keep my initial rating.

---

### Meta-Review · Area_Chair_SKvd · 2024-12-19

**Metareview:**

Under faithfulness violations certain pairwise relations may look independent, and dependence may only become visible when looking at the relation to a set of variables. The authors formalize this notion in an attempt to relax the notorious faithfulness assumption for causal discovery under the causally sufficient setting, i.e., no unobserved confounders.

**Additional Comments On Reviewer Discussion:**

The reviewers provided good feedback ranging from several baselines that are not even cited in the paper, to experiment suggestions. I agree with those. The authors did not submit a rebuttal. I am interpreting this as them admitting that the paper is not ready for publication in its current form. I would recommend resubmission after the reviewer comments are incorporated.

---

### Decision · Program_Chairs · 2025-01-22

Reject